# Melatonin Preserves Fluidity in Cell and Mitochondrial Membranes against Hepatic Ischemia–Reperfusion

**DOI:** 10.3390/biomedicines11071940

**Published:** 2023-07-08

**Authors:** Eduardo Esteban-Zubero, Laura López-Pingarrón, José Manuel Ramírez, Marcos César Reyes-Gonzales, Francisco Javier Azúa-Romeo, Marisol Soria-Aznar, Ahmad Agil, José Joaquín García

**Affiliations:** 1Department of Pharmacology, Physiology and Legal and Forensic Medicine, Faculty of Medicine, University of Zaragoza, 50009 Zaragoza, Spain; eezubero@gmail.com (E.E.-Z.); mreyesg@unizar.es (M.C.R.-G.); msoria@unizar.es (M.S.-A.); 2Department of Surgery, University of Zaragoza, 50009 Zaragoza, Spain; jramirez@unizar.es; 3Department of Human Anatomy and Histology, Faculty of Medicine, University of Zaragoza, 50009 Zaragoza, Spain; jazua@unizar.es; 4Department of Pharmacology, Faculty of Medicine, University of Granada, 18071 Granada, Spain; aagil@ugr.es

**Keywords:** ischemia–reperfusion injury, melatonin, antioxidant, membrane fluidity, lipid peroxidation, carbonyl content, mitochondria

## Abstract

We evaluated the in vivo effects of melatonin treatment on oxidative damage in the liver in an experimental model of ischemia–reperfusion. A total of 37 male Sprague-Dawley rats were randomly divided into four groups: control, ischemia, ischemia + reperfusion, and ischemia + reperfusion + melatonin. Hepatic ischemia was maintained for 20 min, and the clamp was removed to initiate vascular reperfusion for 30 min. Melatonin (50 mg/kg body weight) was intraperitoneally administered. Fluidity was measured by polarization changes in 1-(4-trimethylammoniumphenyl)-6-phenyl-1,3,5-hexatriene-*p*-toluene sulfonate). After 20 min of ischemia, no significant changes were observed in cell and mitochondrial membrane fluidity levels, lipid peroxidation, and protein carbonylation. However, after 30 min of reperfusion, membrane fluidity decreased compared to controls. Increases in lipid and protein oxidation were also seen in hepatic homogenates of animals exposed to reperfusion. Melatonin injected 30 min before ischemia and reperfusion fully prevented membrane rigidity and both lipid and protein oxidation. Livers from ischemia–reperfusion showed histopathological alterations and positive labeling with antibodies to oxidized lipids and proteins. Melatonin reduced the severity of these morphological changes and protected against in vivo ischemia–reperfusion-induced toxicity in the liver. Therefore, melatonin might be a candidate for co-treatment for patients with hepatic vascular occlusion followed by reperfusion.

## 1. Introduction

Hepatic ischemia–reperfusion injury (IRI) is a pathological phenomenon characterized by acute restriction of the blood supply to the liver, followed by the restoration of vascular perfusion. This process may be involved in the physiopathology of numerous diseases, including myocardial infarction, stroke, acute organ injury, or systemic shock, and in surgical procedures, such as organ transplantation. Ischemia results in a severe metabolic imbalance between supply and demand. This imbalance causes hypoxia, which leads to partial or complete loss of hepatic function [1]. During ischemia, parenchymal cell death occurs because of localized cellular metabolic disturbances resulting from lack of oxygen, glycogen consumption, ATP depletion and degradation into its metabolites, and conversion of xanthine oxidase to xanthine dehydrogenase.

The primary function of mitochondria is to generate ATP. During this process, electrons are captured and transported to oxygen to form water. Unfortunately, some electrons inevitably leak out of the electron transport chain (ETC) and form free radicals, mainly the superoxide anion radical (O_2_^•−^). These radicals can be dismutated to form hydrogen peroxide (H_2_O_2_), which can diffuse to other cellular compartments and cause oxidative stress [2]. Paradoxically, reperfusion leads to a burst of free radicals from the mitochondria [1,3]; subsequently, the exacerbated oxidative stress causes more damage to the ETC, which then leads to more electron leakage and free radical production [4,5,6]. Hypoxanthine, accumulated during ischemia, is converted into reactive oxygen species (ROS), which promotes lipid peroxidation [7]. ROS also activates leukocytes and induces chemotaxis, increases leukocyte adhesion molecules, and cytokine gene expression. As a result, tumor necrosis factor-alpha (TNF-α), as well as interleukins (ILs), are secreted, which increases its levels because of its ability to stimulate Kupffer cells. Lipid peroxidation (LPO) determines a massive overload of Ca^2+^ and Na^+^ intracellular concentrations. Moreover, in sinusoidal endothelial cells and hepatocytes, it leads to discharges of cytochrome C from the mitochondria into the cytoplasm because of cell membrane permeability disruption [8]. Ca^2+^ overload thus results in the opening of the mitochondrial permeability transition pores [9]. Finally, whereas ischemia reduces nitric oxide (NO•) levels, reperfusion generates large amounts of reactive nitrogen species (RNS) because of the upregulation of NOS in all liver cells in response to inflammatory mediators [10,11,12].

Melatonin (N-acetyl-5-methoxytryptamine) synthesis occurs in the pineal gland, gastrointestinal tract, airway epithelium, pancreas, adrenal glands, thyroid gland, thymus, urogenital tract, placenta, mast cells, natural killer cells, eosinophilic leukocytes, platelets, and endothelial cells [13]. It is well known that melatonin plays a pivotal role in controlling circadian rhythms [14,15]. It has also been firmly established that indoleamines can directly scavenge hydroxyl radicals (•OH) [16], H_2_O_2_ [17], and O_2_^•−^ [18]. Melatonin also functions as an indirect antioxidant, due to its ability to stimulate the expression and activity of antioxidant enzymes that eliminate ROS [19]. A prominent feature of melatonin is that when it interacts with •OH, it initiates a beneficial scavenging cascade reaction. This reaction produces the metabolites cyclic 3-hydroxymelatonin, N_1_-acetyl-N_2_-formyl-5-methoxykynuramine (AFMK), and N_1_-acetyl-5-methoxykynuramine (AMK), which also contribute to the neutralization of free radicals [20]. Therefore, a single molecule of melatonin could potentially scavenge ten or more ROS molecules.

Several studies have provided evidence that melatonin could protect mitochondria from oxidative stress due to different toxins, including ischemia–reperfusion-induced ROS [21,22]; ethanol [23]; ruthenium red [24]; lipopolysaccharide [25]; and arsenite [26]. Moreover, it has been shown that melatonin upregulates the activity of all four complexes in the ETC under ischemia–reperfusion conditions, which might reduce ROS production [27,28]. Melatonin also preserves the mitochondrial membrane potential (Δψ), which is important for ATP generation and for maintaining full mitochondrial function [29]. It has been shown that indoleamine also protects cardiolipin from oxidation [30]. Cardiolipin binds and stabilizes the closed form of the mitochondrial ATP/ADP carrier. However, the oxidation of cardiolipin causes a modification in the ATP/ADP carrier configuration, which stabilizes its open form; this conformation induces depolarization, alterations in Δψ, and mitochondrial swelling, which result in cellular apoptosis. Melatonin increases the activity of mitochondrial uncoupling proteins [31]. These proteins have beneficial effects on mitochondrial function, including maintaining the Δψ, accelerating electron transport, and reducing ROS formation [32]. Finally, indoleamine enhances mitophagy and improves mitochondrial biogenesis [33].

The preservation of mitochondrial functional integrity is essential for a healthy cell. Accordingly, mitochondrial dysfunction has been implicated in numerous pathological processes, including ischemia–reperfusion. Evidence has suggested that excessive generation of ROS in eukaryotic cells may be due to impairments in the ETC [34,35]. Although cells possess several mechanisms for ROS detoxification, an uncontrolled increase in ROS leads to chain reactions mediated by free radicals interacting indiscriminately with target lipids, proteins, and DNA.

The most abundant constituents of biological membranes are phospholipids and proteins. The lipid peroxidation process in these membranes forms peroxyl radicals (ROO•), which are sufficiently toxic to propagate an oxidative reaction, with adjacent phospholipids as substrates. Then, lipid peroxidation causes the loss of bilayer fluidity and, eventually, the cell membrane breakdown.

In the present study, we investigated the ability of melatonin to resist IRI-induced oxidative stress in rats. Because membrane physical properties modulate numerous cellular functions, cell membrane fluidity was used as an index of IRI and associated cell damage. The tissue concentrations of malonyldialdehyde (MDA) plus 4-hydroxyalkenals (4-HDA) and carbonyl content were used as biochemical markers of lipid and protein oxidation, respectively. Since mitochondria are considered the largest source of ROS in cells and mitochondria harbored higher melatonin concentrations than other organelles and other subcellular locations [36], we also investigated the protective effects of melatonin in maintaining mitochondrial membrane fluidity against ischemia–reperfusion in the liver.

## 2. Materials and Methods

### 2.1. Animals

The handling and animal procedures were carried out in strict accordance with the recommendations of the European Community Committee (2010/63/EU) for the care and use of laboratory animals. Procedures were approved by the University of Zaragoza Ethics Committee for Animal Experiments (PI036/09). A total of 37 male Sprague-Dawley rats weighing 225–250 g were purchased from Harlan-Ibérica (Barcelona, Spain) and housed in Plexiglas cages, two animals per cage, in a windowless room with automatically regulated temperature (22 ± 2 °C) and lighting (12 h light/12 h dark). Animals received standard laboratory chow and water *ad libitum*. The rats were randomly divided into four groups as follows: control, *n* = 8; ischemia, *n* = 10; ischemia + reperfusion, *n* = 9; and melatonin + ischemia + reperfusion, *n* = 10.

### 2.2. Experimental Model and Sample Collection

After acclimation for 2 weeks, animals were fasted overnight prior to surgery, having only free access to water. Animals were anesthetized with intraperitoneal administration of pentobarbital sodium (50 mg/kg body weight) to ensure its rapid availability in the hepatic tissue. A midline laparotomy was performed, and the liver was gently mobilized. In accordance with the Pringle maneuver, an atraumatic microvascular clamp was used to completely block the portal vein, hepatic artery, and bile duct [37]. Hepatic ischemia was maintained for 20 min, and the clamp was removed to initiate vascular reperfusion for 30 min. A simulated surgery, without the vascular occlusion, was performed in the control group. Rat body temperature was maintained at 37 °C throughout the surgical procedure. Thirty minutes before ischemia initiation, melatonin was intraperitoneally administered at a dose of 50 mg/kg body weight. Non–melatonin-treated groups received an intraperitoneal injection of the same volume of saline. Melatonin was prepared immediately before use, by dissolving it in absolute ethanol and saline, with an ethanol concentration of 1% in the final mixture. Rats were sacrificed by decapitation, and the livers were removed quickly, washed in saline (4 °C), and stored for a maximum of 1 month at −80 °C until use.

### 2.3. Membrane Isolation and Fluidity Measurements

Cell membrane fractions were isolated using the differential centrifugation method described by Aranda et al. with minor modifications [38]. Livers were homogenized in 1/10 *w*/*v* in 140 mM KCl/20 mM 4-(2-hydroxyethyl)-1-piperazineethanesulfonic acid buffer (HEPES) (pH = 7.4) and then centrifuged at 1000× *g* for 10 min at 4 °C to clear the fluid of cell debris and nuclear material. The resulting supernatant was centrifuged at 50,000× *g* for 20 min at 4 °C. The pellet, containing the cell membranes and mitochondria, was re-suspended in the same buffer and centrifuged at 10,000× *g* for 15 min to separate the mitochondrial fraction: the resulting supernatant, containing the cell membranes, was centrifuged again at 50,000× *g* for 20 min at 4 °C; and the pellet, containing the mitochondrial membranes, was re-suspended in 1/10 *v*/*v* HEPES buffer and re-centrifuged at 10,000× *g* for 15 min. Then, separately, cell and mitochondrial membranes were re-suspended 1/10 *v*/*v* using the same buffer and re-centrifuged at 50,000 or 10,000× *g* for 15 or 20 min, respectively. Finally, both pellets were re-suspended 1/1 *v*/*v* in 50 mM tris(hydroxymethyl)aminomethane (TRIS) (pH = 7.4) and stored at −80 °C until used in the assay.

Membrane fluidity was monitored using fluorescence spectroscopy and 1-(4-trimethylammoniumphenyl)-6-phenyl-1,3,5-hexatriene-*p*-toluene sulfonate (TMA-DPH) as a probe. Its incorporation into the membrane and the determination of membrane fluidity were carried out according to methods described previously [39]. Separately, cell and mitochondrial membranes (0.5 mg protein/mL) were re-suspended in 50 mM TRIS (3 mL final volume), mixed vigorously with TMA-DPH (66.7 nM), and incubated for 30 min at 37 °C. Fluorescence measurements were performed in a Perkin-Elmer^®^ LS-55 Fluorescence Spectrometer equipped with a circulatory water bath to maintain the temperature at 22 ± 0.1 °C. Excitation and emission wavelengths of 360 and 430 nm, respectively, were used. The emission intensity of vertically polarized light was detected by an analyzer oriented parallel (IV_V_) or perpendicular (IV_H_) to the excitation plane. A correction factor (G) for the optical system was used. Polarization (P) was calculated by the following equation:$$P=\frac{{\mathrm{IV}}_{V}-{\mathrm{GIV}}_{H}}{{\mathrm{IV}}_{V}+{\mathrm{GIV}}_{H}}$$

Because an inverse relationship exists between fluidity and polarization, membrane fluidity was expressed as 1/P. This value was calculated as the arithmetic mean of 30 independent measurements. Protein concentrations were determined by the Bradford method using bovine serum albumin as the standard [40].

### 2.4. Assay of Lipid Peroxidation

MDA + 4-HDA were measured in the liver homogenates. Briefly, tissues were homogenized in 50 mM TRIS (pH = 7.4) buffer. A total of 200 µL of the suspension was mixed with 650 µL of a methanol/acetonitrile (1/3, *v*/*v*) solution containing N-methyl-2 phenyl-indole. After the addition of 150 µL of methanesulfonic acid, incubation was carried out at 45 °C for 40 min. MDA + 4-HDA concentrations were measured with a spectrophotometer at 586 nm using 4-hydroxy-nonenal as the standard. The levels of LPO in the homogenates were expressed as nanomoles of MDA + 4-HDA per mg of protein.

### 2.5. Protein Carbonylation

The carbonyl content of proteins was quantified in the liver homogenates according to the method described by Levine et al. [41]. In this assay, 2,4-dinitrophenylhydrazine (DNPH) is a chromogen that interacts with the carbonyl groups of the oxidatively damaged proteins, reaching a maximal absorbance of 375 nm. Protein carbonyl content, expressed in nanomoles per mg of protein, was estimated with the molar absorption coefficient of 22,000/M/cm for DNPH derivatives. Guanidine-HCl solution was used as a blank.

### 2.6. Immunohistochemical Studies

Pieces of liver tissue were fixed in 10% formalin buffered with phosphate solution (0.1 M, pH 7.4) for 48 h at room temperature. Fragments of liver were washed in phosphate buffer and dehydrated in graded concentrations of ethanol (70%, 80%, 90%, and 100%). The fragments were then embedded in paraffin and subsequently sectioned (4 μm thickness). Hematoxylin and eosin staining were performed according to the standard procedure to evaluate hepatic histopathology.

Immunohistochemistry was carried out using primary mouse antibody to MDA (1 µg/mL, JaiCA, Fukuroi, Shizuoka, Japan, MMD-030) and primary mouse antibody antidityrosine (MDT) (1:500, JaiCA, MDT-020P), and the specific biotinylated secondary antibodies (Leica Biosystems, Barcelona, Spain, Novolink MinPolymer^®^ DS RE7290-CE). Immunohistochemical labeling was revealed using the peroxidase diaminobenzidine (DAB) method, with counterstaining with hematoxylin for qualitative identification of antigens by light microscopy. After mounting by means of Eukitt^®^, sections were examined with an OLYMPUS CX51 microscope, and images were scanned under equal light conditions using the Cell-D program (Olympus, Hicksville, NY, USA). Two histologists, unaware of the treatments applied during the experiments, examined the stained slides independently. Each sample was observed at 100–400× magnification, and every specimen was analyzed containing a centrilobular vein.

### 2.7. Statistical Analyses

Each variable was defined by the arithmetic mean as a measure of central tendency and by the standard error as a measure of dispersion. Subsequently, once the variables were verified to follow a normal distribution by the Kolmogorov–Smirnov test, an inferential study was carried out using the two-tailed non-paired student’s t-test to determine statistical differences among the groups. A *p* value of <0.05 was considered significant. Statistical analysis and elaboration of the graphs were performed using Statistical Product and Service Solution software, version 22.0, and Sigma Plot Software, version 13.0, respectively.

## 3. Results

### 3.1. Cell and Mitochondrial Membrane Fluidity

After 20 min of ischemia, the cell (Figure 1a) and mitochondrial (Figure 1b) membrane fluidity levels (1/P = 3.83 ± 0.06 and 3.99 ± 0.12, respectively) were similar to those measured at baseline in control rats (1/P = 3.79 ± 0.03 and 4.09 ± 0.12, respectively). However, after 30 min of vascular reperfusion, the cell and mitochondrial membrane fluidity significantly decreased (1/P = 3.23 ± 0.08 and 3.63 ± 0.10, respectively; *p* ≤ 0.0001), which indicated that cell and mitochondrial membrane rigidity increased, compared to the control group. When rats were treated with melatonin (1/P = 3.83 ± 0.04 and 4.04 ± 0.10) 30 min before the induction of ischemia, the indoleamine prevented (*p* ≤ 0.0001) both cell (Figure 1a) and mitochondrial (Figure 1b) cell rigidity by 100% and 87.91%, respectively. In fact, melatonin-treated rats had hepatic and mitochondrial cell membrane fluidity levels statistically similar to the fluidity observed in the membranes of control animals.

### 3.2. Biochemical Indices of Oxidative Stress

#### 3.2.1. Lipid Peroxidation

Lipid peroxidation is an auto-oxidative process in which free radicals interact with the fatty acids present in cell membranes to produce a wide variety of oxidation products. Among them, MDA and 4-HDA have been widely used for the last four decades as convenient biomarkers for lipid peroxidation [42].

Figure 2 summarizes the results of lipid peroxidation. The concentrations of MDA + 4-HDA in the group of control animals was 0.36 ± 0.04 nmol/mg protein, very similar to those obtained in rats subjected to 20 min of ischemia (0.33 ± 0.03 nmol/mg protein). After 30 min of vascular reperfusion, an increase in lipid oxidative damage was observed (0.73 ± 0.06 nmol/mg protein), which was completely avoided when, in addition to vascular reperfusion, melatonin was previously administered (0.36 ± 0.02 nmol/mg protein). Lipid peroxidation induced by reperfusion was statistically significant (*p* ≤ 0.001 vs. control group and *p* ≤ 0.0001 vs. ischemia and ischemia–reperfusion treated with melatonin).

#### 3.2.2. Protein Carbonylation

Protein carbonylation is an index of protein oxidative damage due to free radicals. The concentration of protein carbonyl residues in control cases and post-ischemia for 20 min were 1.69 ± 0.25 and 1.75 ± 0.25 nmol/mg protein, respectively. After 30 min of reperfusion, there was a marked increase (4.08 ± 0.24) compared to the other groups (*p* ≤ 0.0001). In addition, the concentrations of protein carbonylation in rats exposed to IRI and co-treated with melatonin (1.84 ± 0.19 nmol/mg protein) decreased (*p* ≤ 0.0001) by 95% relative to levels measured in the liver of rats exposed to IRI, with no significant differences vs. control rats. Figure 3 illustrates these results.

### 3.3. Immunohistochemical Features

The protective effect of melatonin against ischemia–reperfusion was further confirmed by the immunohistochemical study using anti-MDA and MDT. We selected these antibodies since MDA is the most abundant individual aldehyde resulting from lipid peroxidation [43], and MDT is a tyrosine dimer derived from tyrosyl radicals and is considered one of the specific biomarkers for protein oxidation [44]. The livers of the control group for both immunohistochemical staining exhibited normal lobular architecture with central veins and radiating hepatic cords (Figure 4a,e). Microscopic examination of the liver after ischemia for 20 min (Figure 4b,f) showed cytoplasmic vacuolization of hepatocytes and discrete signs of oxidative stress at the level of the hepatocyte cytoplasm in the vicinity of the centrilobular vein, as well as weak leukocyte infiltration. After reperfusion, they displayed hyperchromatic hepatocyte nuclei as a sign of nuclear hyperactivity and, an increase in the labeling of anti-MDA (Figure 4c) at the level of cytoplasm and membranes of the hepatocytes that are close to the centrilobular vein, which indicates intense lipid peroxidation in these areas. Figure 4g also shows signs of oxidative stress in the proteins in the region near the centrilobular vein, although less marked than in the lipid oxidation staining. Finally, in the livers of the rats treated with melatonin, the signs of oxidative stress in lipids (Figure 4d) and proteins (Figure 4h) due to ischemia–reperfusion were lower at the level of the centrilobular vein, as well as in the cytoplasms and hepatic sinusoids.

## 4. Discussion

Ischemia–reperfusion initiates a complex cascade of molecular events that contribute differently to the overall injury. These events include mitochondrial dysfunction, acidosis, calcium release from intracellular stores, apoptosis, failure of capillary perfusion and subsequent endothelial damage, and the release of proinflammatory mediators and transcription factors [45,46,47].

There is a consensus among researchers that the ischemic period primes the tissue for subsequent damage upon reperfusion. According to our results, ischemia did not disturb oxidative stress markers, as previously reported [48,49]. In the early phase of ischemic injury, acidosis affords protection for hepatocytes against cell necrosis. However, when ischemia injury persists, this protective mechanism is insufficient to avoid the cascade of events that finally leads to parenchymal cell death [50]. In addition, ATP depletion occurs, as does its degradation into metabolites such as adenosine, inosine, and hypoxanthine, especially at the mitochondrial level [51]. NO levels also fall because of the diminished activity of nitric oxide synthase (NOS) resulting from the lack of oxygen generated during the ischemia process [52].

The present study demonstrated that melatonin pretreatment significantly attenuates LPO triggered by experimental IRI in the liver. Melatonin antioxidant activity against oxidative stress has been well documented in a variety of in vitro and in vivo experimental conditions. The indoleamine prevents ROS formation during reperfusion because it preserves ATP, ameliorates the imbalanced expression of the vascular stress genes during hepatic IRI, and increases the activity of antioxidant enzymes [53,54,55,56,57]. In addition, our results show melatonin was also effective in reducing the carbonyl content. Protein carbonylation is a type of protein oxidation that ROS can promote. These species arise from several chain reactions initiated as a response to the oxidative decomposition of polyunsaturated fatty acids [58]. Similar findings have been previously reported in a testicular IRI model developed in rats after 1, 3, and 24 h of reperfusion [59].

The fluid properties of biological membranes are determined mainly by the length and saturation of the fatty acid chains in phospholipid molecules located in the lipid bilayer [60]. Polyunsaturated fatty acids exhibit the highest sensitivity to free radical attack in biological membranes. Moreover, membrane lipids can undergo cross-linking, which might limit phospholipid mobility and contribute to rigidity [61]. We previously showed that reperfusion during pancreas transplantation reduced the fluidity of cell membranes isolated from pancreatic allografts [62]. We also showed that after reperfusion, membrane rigidity was prevented by replacing the University of Wisconsin preservation solution (UW), the universally accepted standard for perfusion and cold storage of abdominal organs, with the lower viscosity, Institute Georges Lopez-1 solution [63]. The Georges Lopez-1 solution contained polyethylene glycol, a polyether compound that is adsorbed to the surface of cell membranes. It preserves or restores the integrity of plasma membranes and reduces lipid peroxidation due to reperfusion during organ transplantation [64].

Based on this evidence, it seems reasonable to be interested in antioxidant molecules, like melatonin, because they might protect mitochondria during ischemia–reperfusion by stabilizing their membranes. Clearly, these effects depend on the accumulation of antioxidant agents in mitochondria. Melatonin is highly concentrated in mitochondria; its levels in mitochondria are roughly 100-fold higher than its plasma levels [65]. Studies have shown that mitochondria can synthesize melatonin from tryptophan; furthermore, melatonin can be catabolized to form the free-radical scavenger, AFMK [33,66].

We previously tested the effects of melatonin in preserving hepatic microsomal membrane fluidity using a well-established in vitro model, and we observed that melatonin protected against lipid peroxidation induced by the addition of FeCl_3_, ADP, and nicotinamide adenine dinucleotide phosphate. Membrane rigidity increased due to lipid peroxidation, but the addition of melatonin reduced both microsomal membrane rigidity and lipid peroxidation [67]. Melatonin was also tested in patients that underwent cardiac surgery that required a cardiopulmonary bypass. This aggressive procedure is frequently complicated by the ischemia–reperfusion sequence syndrome. After the cardiopulmonary bypass, patients exhibited increases in thiobarbituric acid-reactive substances (TBARS) and erythrocyte membrane rigidity compared to the reference sample. Treatment with melatonin prevented both the change in TBARS and the erythrocyte membrane rigidity [68].

To our knowledge, the present study was the first to provide evidence that an injury due to vascular reperfusion was characterized by a reduction in the fluidity of mitochondrial membranes. It has been proposed that ROS and elevated calcium concentrations might damage the mitochondrial membrane structure [69]. The exacerbated oxidative stress levels during reperfusion were closely correlated with the rigidity of mitochondrial membranes isolated from these hepatic tissues. This correlation suggested a potential cause/effect relationship between these phenomena [61].

## 5. Conclusions

IRI is implicated in a cascade of oxidative stress damage arising from an increase in inflammatory mediators, leukocyte recruitment, ATP depletion, and the subsequent generation of ROS and RNS. Consequently, LPO destroys lipids and contributes to cell edema, likely because of disruption of normal cell membrane fluidity. In addition, a process of protein oxidation is involved in promoting apoptosis and cell death.

Our results indicate that melatonin is effective in preserving both cell and mitochondrial membrane fluidity and mitigating the increased MDA and carbonyl content. This evidence strongly suggests the ability of melatonin to preserve optimal levels of fluidity in biological membranes and must be considered another important mechanism by which melatonin plays a beneficial role in protecting against ischemia–reperfusion injury.

These results are consistent with other studies that showed that melatonin administration could reduce mitochondrial dysfunction, lipid peroxidation, and protein oxidation. Given its potent antioxidant effects, melatonin might be useful during IRI for preventive and therapeutic purposes.

## Figures and Tables

**Figure 1 biomedicines-11-01940-f001:**
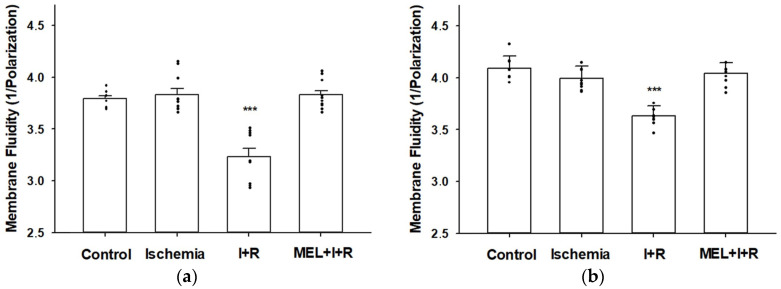
Effect of melatonin (MEL) administration at a dose of 50 mg/kg body weight on cell (**a**) and mitochondrial (**b**) membrane fluidity (expressed as 1/Polarization) in rat livers exposed to ischemia (I) and reperfusion (R). Maximum levels of rigidity were reached after 30 min of vascular reperfusion. Melatonin completely prevented the rigidity due to reperfusion. Data are expressed as means ± standard errors of 8–10 rats per group. *** *p* < 0.0001 vs. control, ischemia and MEL + I + R.

**Figure 2 biomedicines-11-01940-f002:**
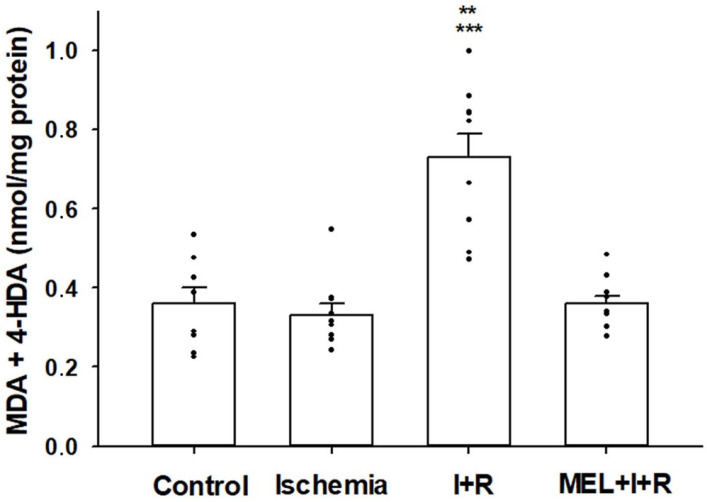
The level of malondialdehyde (MDA) and 4-hydroxyalkenals (4-HDA) concentrations in hepatic homogenates of rats exposed or not to ischemia (I) or ischemia + reperfusion (I + R), or treated with melatonin (MEL) 30 min before I + R. Bars represent means ± standard errors of 8–10 animals per group. ** *p* < 0.001 vs. control rats. *** *p* < 0.0001 vs. ischemia and MEL + I + R groups.

**Figure 3 biomedicines-11-01940-f003:**
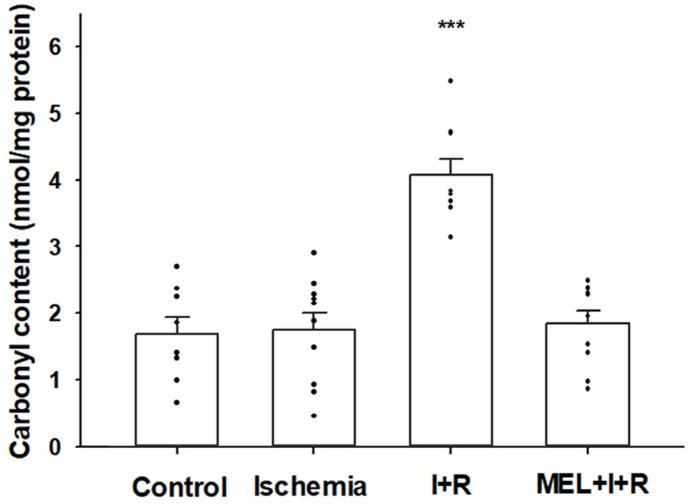
Protein carbonylation in the liver homogenates from sham-operated rats (controls), rats exposed to ischemia (I) for 20 min, and rats exposed to ischemia + 30 min of reperfusion (R) or treated with melatonin (MEL) in combination with I + R. Data are expressed as means ± standard errors of 8–10 animals per group. *** *p* < 0.0001 vs. control, ischemia, or MEL + I + R groups.

**Figure 4 biomedicines-11-01940-f004:**
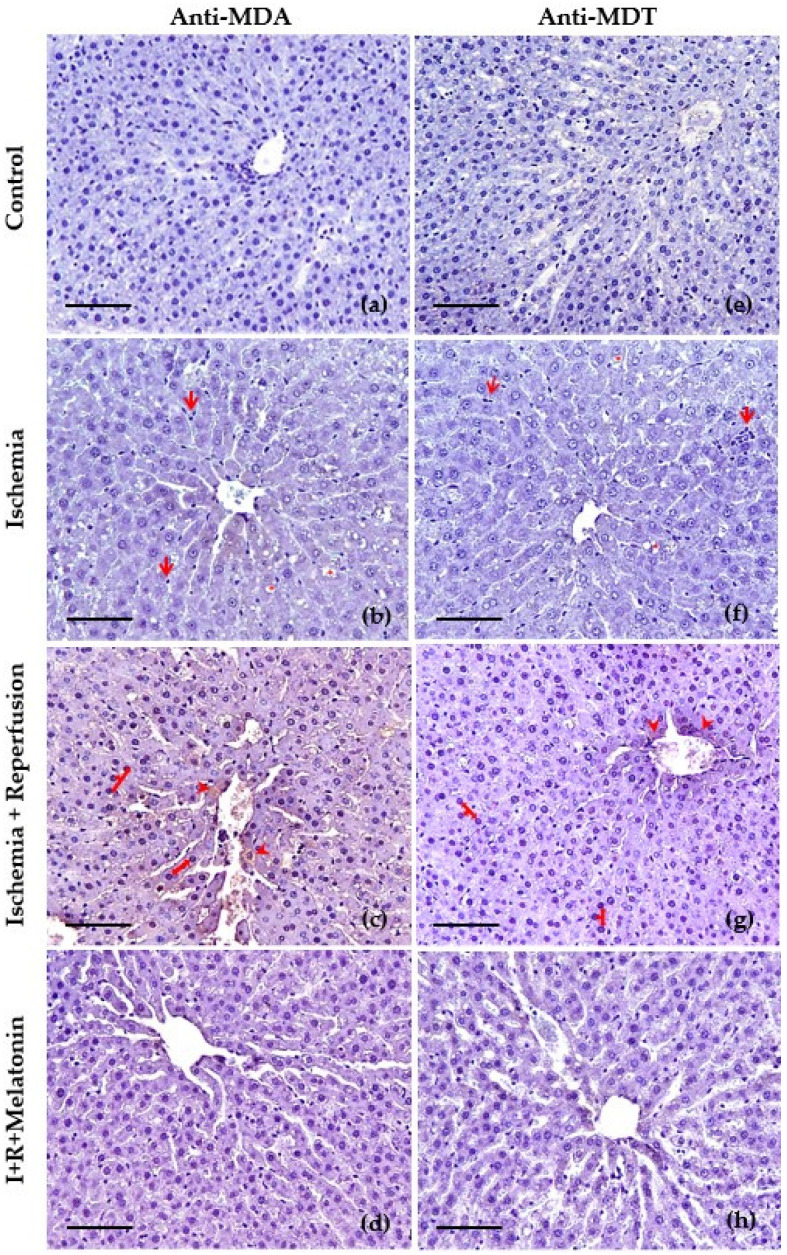
Slides of hepatic tissues at the level of centrilobular vein obtained from control, exposed to ischemia (I), after reperfusion (R), and co-treated with melatonin (I + R + Melatonin). Images were captured at 200×. Scale bar: 100 μm. Immunohistochemical study using primary antibody to malondialdehyde (MDA) (**a**–**d**) and anti-dityrosine (MDT) (**e**–**h**). The main changes in ischemia (**b**,**f**) were hepatic vacuolization (red asterisks *), and leukocyte infiltration (red arrows), whereas the reperfusion indicated mainly hyperchromatic hepatocyte nuclei (red arrows) and oxidative damage (DAB+) (red arrowheads) especially to lipids (**c**) and to a lesser degree to proteins (**g**), in the proximity of the centrilobular vein (**c**,**g**), and in the hepatic cords (**c**). The concurrent administration of melatonin reduced the severity of these histopathological findings (**d**,**h**).

## Data Availability

The data presented in this study are available on reasonable request from the corresponding author.

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
