# Peer review of "Melatonin Preserves Fluidity in Cell and Mitochondrial Membranes against Hepatic Ischemia–Reperfusion"

_biomedicines, 2023, doi:10.3390/biomedicines11071940_

Round 1

Reviewer 1 Report

In the current study the authors have explored the antioxidant effect of melatonin (50mg/Kg body wt) in rescuing Ischemia reperfusion (IR) injury-induced hepatocellular damage. The authors took advantage of in vivo hepatic IR model using SD rats  housed under standard control conditions and appropriate assortment of the animals have been made based on the experimental design. Relevant ethical approval and IACUC protocols have been followed throughout the study. Here by using differential centrifugation method to isolate the cell membrane and the mitochondrial membranes, and by employing assays to determine membrane fluidity, lipid peroxidation, protein carbonylation the authors have provided evidence that melatonin was effective in rescuing the IR-induced hepatocellular damage by restoring cellular and mitochondrial membrane fluidity, reversing IR-induced lipid peroxidation and protein carbonylation. Using immunohistochemical study of the formalin fixed paraffin embedded tissue sections and using anti-MDA and –dityrosine antibodies the authors attempted to provide evidence that the IR-induced cellular damage showing cytoplasmic vacuolization of hepatocytes,  and other pathological signs of oxidative cellular danages have been partially restored by melatonin pretreatment. In conclusion, the study determined melatonin as a possible further research to explore melatonin as preventive and therapeutic approaches to treat IR-induced hepatocellular damage.

Overall, this is a well-written manuscript. Appropriately controlled experimental protocols have been used and apparently the conclusion seems supported by the experimental data.

I have a few comments and concerns regarding the study.

1.       The discussion section mostly describes the studies performed by other groups or the studies already published in this subject area by the present authors and discussion on the results obtained in the current study is limited. I would recommend moving the paragraph (p8, line 288-301) to the introduction section. Similarly lines 352-367 discusses results derived from other studies and mostly speculative. No data supporting this section has been presented in the current study.

2.       The conclusion of the study mentions absence of toxicity of melatonin in the experimental dose but no toxicity studies have been performed. This is an overstatement and the authors are recommended to provide evidence-based conclusion of the study.

3.        The study is limited with low sample size in each experimental group. Did the authors perform any power analysis to determine minimum number of animals required in each group to see an effect due to experimental maneuver with confidence?

4.       Given the low sample size the statistical analysis ran on each data set appears too precise with very less dispersion of raw data points. The authors are requested to show all the bar graphs presented in the manuscript with box plot showing the raw data points to clearly show the dispersion of the data as collected.

5.       The IHC data (Fig  4) do not have any magnification parameter or the scale bar annotated. Without those information the data seem incomplete. Also appropriately mark with colored arrowhead to indicate the IR-induced pathological tissue damage and protective effect of melatonin at the experimental dose.

6.       Are the anti-MDA and anti-MDT antibodies used in the IHC study validated against rat species?  The IHC images appear saturated with hematoxylin and bluing and the overall DAB signal is poor and nothing convincingly could be extracted from those images as to be definitive on either Ir-induced cellular damage or the effect of melatonin as a protective agent. Where are the HE and negatively stained IHC images?

Author Response

Dear Reviewer 1:

Please, find attached the resubmission of the revised manuscript biomedicines-2445067 “Melatonin preserves fluidity in cell and mitochondrial membranes
against hepatic ischemia-reperfusion”. The point-by-point responses are presented below.

We would like to thank for your insightful comments. We have made changes throughout the manuscript and hope that this revision will improve the overall quality of the manuscript.

Point 1: The discussion section mostly describes the studies performed by other groups or the studies already published in this subject area by the present authors and discussion on the results obtained in the current study is limited. I would recommend moving the paragraph (p8, line 288-301) to the introduction section. Similarly lines 352-367 discusses results derived from other studies and mostly speculative. No data supporting this section has been presented in the current study.

Response: Both of the indicated paragraphs have been moved to the introduction section.

Point 2: The conclusion of the study mentions absence of toxicity of melatonin in the experimental dose but no toxicity studies have been performed. This is an overstatement and the authors are recommended to provide evidence-based conclusion of the study.

Response: I thank you for the observation, in the study we have not done studies on the toxicity of melatonin so it should not be included in the conclusions. We have eliminated it.

Point 3: The study is limited with low sample size in each experimental group. Did the authors perform any power analysis to determine minimum number of animals required in each group to see an effect due to experimental maneuver with confidence?

Response: Sample size was calculated for membrane fluidity measurements, MDA+4-HDA concentrations and protein carbonylation using arithmetic mean, according to the results previously obtained for control rats (https://doi.org/10.1016/s0014-5793(97)00447-x; https://doi.org/10.1111/j.1600-079x.2010.00769.x; https://doi.org/10.1002/1097-4644(20010315)80:4%3C461::aid-jcb1000%3E3.0.co;2-p;) and the equation n = (Zα/2﮲σ/e)2, where “n” is simple size, “Zα/2” 1.96 for a IC95%, and “e” is interval semi-amplitude, determining that data from 6-8 rats in each group were required. These sample sizes were confirmed using GPower 3.1 software.

Point 4: Given the low sample size the statistical analysis ran on each data set appears too precise with very less dispersion of raw data points. The authors are requested to show all the bar graphs presented in the manuscript with box plot showing the raw data points to clearly show the dispersion of the data as collected.

Response: Following your instructions we have changed the figures showing the raw data points in Figures 1, 2 and 3.

Point 5: The IHC data (Fig 4) do not have any magnification parameter or the scale bar annotated. Without those information the data seem incomplete. Also appropriately mark with colored arrowhead to indicate the IR-induced pathological tissue damage and protective effect of melatonin at the experimental dose.

Response: We appreciate your comment and we have added the magnification parameter in the Figure 4 legend, and used mark with colored arrowhead and arrows to indicate the pathological findings.

Figure 4. Slides of hepatic tissues at the level of centrolobular vein obtained from control, exposed to ischemia (I), after reperfusion (R) and co-treated with melatonin (I+R+Melatonin). Images were captured at 200x. Immunohistochemical study using primary antibody to malondialdehyde (MDA) (a-d) and anti-dityrosine (MDT) (e-h). The main changes of ischemia (b and f) were hepatic vacuolization (red asterisks*), and leukocyte infiltration (red arrows), whereas the reperfusion indicated mainly hyperchromatic hepatocyte nuclei (red arrows) and oxidative damage (DAB+) (red arrowheads) specially to lipids (c) and to a lesser degree to proteins (g), in the proximity of the centrolobular vein (c and g), and in the hepatic cords (c). The concurrent administration of melatonin reduced the severity of these histopathological findings (d and h).

Point 6: Are the anti-MDA and anti-MDT antibodies used in the IHC study validated against rat species? The IHC images appear saturated with hematoxylin and bluing and the overall DAB signal is poor and nothing convincingly could be extracted from those images as to be definitive on either Ir-induced cellular damage or the effect of melatonin as a protective agent. Where are the HE and negatively stained IHC images?

Responses:

Both antibodies used are validated against rat. A recent technical report (https://doi.org/10.1293/tox.2021-0006) recommends the use of JAICA's anti-MDA especially for the rat and anti-MDT for rat, mouse and dog. Several studies also have used in rats the anti-MDA (https://doi.org/10.1371/journal.pone.0243660) and the anti MDT (https://doi.org/10.1080/10715760500053461; https://doi.org/10.1292/jvms.10-0371; https://doi.org/10.1292/jvms.11-0088).

If the images are saturated with hematoxylin, this is to highlight cell nuclei, although to the detriment of DAB labeling. In fact, although the DAB signal may be appear poor, there is a change observed at the level of centrolobular vein and in the hepatic cords, most of all in the Figure “4c” which shows the oxidative damage to lipids.

We have not provided the hematoxylin and eosin images, nor the negative control for not overloading the manuscript, and, mainly, because they do not provide relevant information on oxidative damage as evidenced by immunohistochemistry. In this sense, we have considered that the control images (4a and 4e) would provide morphological details, with which to be able to compare the rest of the study groups. The hematoxylin was used as a counterstain for immunohistochemistry study

Reviewer 2 Report

The manuscript by Esteban-Zuber et al, shows protective roles of melatonin on oxidative damage during hepatic ischemia-reperfusion by preventing membrane rigidity as well as lipid and protein oxidation. According to the authors this is the first study to show that vascular reperfusion injury reduces mitochondrial membrane fluidity. Although, it’s an interesting in-vivo study showing beneficial roles of melatonin as a possible candidate for co-treatment of hepatic vascular occlusion followed by reperfusion. However, beneficial roles of melatonin in hepatic ischemia/reperfusion injury have already been reported in past, thus making this study an addition to previously described roles of melatonin.  Here are few of my suggestions that might help improve this manuscript:

1.       In Figure 1, which statistical test were performed? The authors reported p-value < 0.0001, however there is only one star on the bars, instead of commonly used three stars.

2.       Page 5, opening statement of “section 3.2.1. Lipid peroxidation” should be focused on why the authors decided to study lipid peroxidation instead of simply stating what figure 2 summaries.

3.       In the “section 3.3 Immunohistochemical features”, the authors stated, “The protective effect of melatonin against ischemia-reperfusion was further confirmed by the immunohistochemical study using antibodies anti MDA and MDT.”. They should also explain in sentence what these proteins are and why it’s important to study them for IR.

4.        For figure 4 the authors mentioned, “Finally, in the livers of the rats treated with melatonin, the signs of oxidative stress in lipids (Figure 4d) and proteins (Figure 4h) due to ischemia-reperfusion were lower at the level of the centrilobular vein, as well as in the cytoplasms and hepatic sinusoids.”.  It should be made clear to the readers which results show reduction in levels of oxidative stress of proteins and which ones are for lipids.

English is fine.

Author Response

Dear Reviewer 2:

Please, find attached the resubmission of the revised manuscript biomedicines-2445067 “Melatonin preserves fluidity in cell and mitochondrial membranes
against hepatic ischemia-reperfusion”.

Our study focuses on assessing the effects of hepatic ischemia reperfusion on biological membranes and whether the antioxidant melatonin can protect them. For this purpose, we studied in cellular and mitochondrial membranes, a functional parameter, their fluidity, and two biochemical indicators of oxidative stress of lipids and proteins, which are the main constituents of biological membranes. The study was completed with two morphological indicators of lipid and protein damage by immunohistochemistry. The main novelties of this manuscript are: 1) Vascular reperfusion in the liver induced cell mitochondrial membrane rigidity; 2) Melatonin administration before ischemia and reperfusion fully prevented membrane rigidity. The point-by-point responses are presented below.

We would like to thank for your insightful comments. We have made changes throughout the manuscript and hope that this revision will improve the overall quality of the manuscript.

Point 1: In Figure 1, which statistical test were performed? The authors reported p-value < 0.0001, however there is only one star on the bars, instead of commonly used three stars.

Response: To determine statistical differences among the groups in Figure 1 we used the two-tailed non-paired student’s t-test. Following your indications we have changed the number of stars used to indicate the probability in Figure 1 (from 1 to 3). We have also modified Figures 2 and 3 to standardize this criterion throughout the manuscript.

Point 2: Page 5, opening statement of “section 3.2.1. Lipid peroxidation” should be focused on why the authors decided to study lipid peroxidation instead of simply stating what figure 2 summaries.

Response: Thank you for your advice. We have added a new sentence to better explain why LPO and MDA+4-HDA were included in the study: “Lipid peroxidation is an autooxidative process to which free radicals interact with the fatty acids present of cell membranes to produce a wide variety of oxidation products. Among them, MDA and 4-HDA have been widely used for the last four decades as a convenient biomarkers for lipid peroxidation [42].”

Point 3: In the “section 3.3 Immunohistochemical features”, the authors stated, “The protective effect of melatonin against ischemia-reperfusion was further confirmed by the immunohistochemical study using antibodies anti MDA and MDT.”. They should also explain in sentence what these proteins are and why it’s important to study them for IR.

Response: We thank you for your comment and include the sentence "We selected these antibodies since MDA is the most abundant individual aldehyde resulting from lipid peroxidation [43], and MDT is a tyrosine dimer derived from tyrosyl radicals and it is considered as one of the specific biomarkers for protein oxidation [44]." in section 3.3 and remove it from section 2.6.

Point 4: For figure 4 the authors mentioned, “Finally, in the livers of the rats treated with melatonin, the signs of oxidative stress in lipids (Figure 4d) and proteins (Figure 4h) due to ischemia-reperfusion were lower at the level of the centrilobular vein, as well as in the cytoplasms and hepatic sinusoids.”. It should be made clear to the readers which results show reduction in levels of oxidative stress of proteins and which ones are for lipids.

Response: We have added several marks in the figure 4 in order to clarify this aspect. The main changes of ischemia (b and f) were hepatic vacuolization (red asterisks*), and leukocyte infiltration (red arrows), whereas the reperfusion indicated mainly oxidative damage (DAB+) (red arrowheads), specially to lipids (c) and to a lesser degree to proteins (g), in the proximity of the centrilobular vein (c and g), and in the hepatic cords (c).

Reviewer 3 Report

Thanks for the opportunity to review the paper titled, "Melatonin preserves fluidity in cell and mitochondrial membranes against hepatic ischemia-reperfusion" by Esteban-Zubero et al. The authors present an interesting topic, but I consider this manuscript not suitable for publication in Biomedicines since the content is very limited as well as the results. The materials and methods section can be greatly improved, there are serious deficiencies in the experimental design, and this directly affects the results. 

Nonetheless, I suggest to authors some feedback suggestions apply to other journals.

-        The introduction is fine but the conclusions are poor with the results reported.

-        Is melatonin injected before surgery or when the animal has already undergone surgery? What clinical sense does it make for a patient to be injected with melatonin half an hour before ischemia if it is unlikely that this event will be known?

-        Authors should explain why the intraperitoneal route is used to inject melatonin and not another route.

-        Specific ethics committee's approval and reference should be indicated.

-        The expression "Figure 1a and 1b illustrate these results." is wrong, it is better to introduce it inside the paragraph. In addition, the results belonging to Figure 1a and Figure 1b are not specified in the text. It is important to note the results concerning each graph.

-        Why is the mean and its standard error represented instead of the standard deviation? The authors should justify this analysis.

-        To support and confirm the lipid peroxidation results, an assay for ROS determination in hepatocytes should be performed.

-        Immunohistochemical assays with specific markers against T-cells (CD3, CD4, CD8), macrophages (like IBA-1) and neutrophils should be performed.

-        What about glycogen synthesis in this tissue? PAS staining should be included to analyze PAS-positive liver cells.

-        Figure 4 is very poorly displayed; it should be included at higher magnification to observe the immunostaining because it is not very conclusive in this way. In addition, it also does not show details such as leukocytic infiltration and increased lipid peroxidation in the sinusoids.

-        All events highlighted by the authors in section 4 should also be marked with arrows or other graphic elements.

Some expressions are translated literally into English from the source language.

Author Response

Dear Reviewer 3:

Please, find attached the resubmission of the revised manuscript biomedicines-2445067 “Melatonin preserves fluidity in cell and mitochondrial membranes
against hepatic ischemia-reperfusion”.

The point-by-point responses are presented below. We would like to thank for your insightful comments. We have made changes throughout the manuscript and hope that this revision will improve the overall quality of the manuscript.

Point 1: The introduction is fine but the conclusions are poor with the results reported.

Response: We thank you for your comment and include in the conclusions the sentence “This evidence strongly suggest the ability of melatonin to preserve optimal levels of fluidity in biological membranes and must be considered another important mechanism by which melatonin plays a beneficial role in protecting against ischemia-reperfusion injury.

Point 2: Is melatonin injected before surgery or when the animal has already undergone surgery? What clinical sense does it make for a patient to be injected with melatonin half an hour before ischemia if it is unlikely that this event will be known?

Response: Melatonin was administered thirty minutes before ischemia initiation. We administered melatonin before ischemia/reperfusion to increase the availability of antioxidants in hepatic tissue and to get cell and mitochondrial membranes in better condition to combat oxidative stress caused by reperfusion. Thus, melatonin preconditioned membranes to prevent the appearance of oxidative injury indicators (MDA-4-HDA and carbonylation for lipids and proteins, respectively).

Point 3: Authors should explain why the intraperitoneal route is used to inject melatonin and not another route.

Response: Thank you for your advice. We have added a new sentence in line 131: “,to ensure its rapid availability in the hepatic tissue.

Point 4: Specific ethics committee's approval and reference should be indicated.

Response: We thank this suggestion. Lines 397-8 included the phrase “Procedures were approved by the University of Zaragoza Ethics Committee for Animal Experiments (PI036/09).” . We also place it in material and methods, lines 120-1.

Point 5: The expression "Figure 1a and 1b illustrate these results." is wrong, it is better to introduce it inside the paragraph. In addition, the results belonging to Figure 1a and Figure 1b are not specified in the text. It is important to note the results concerning each graph.

Response: Following your recommendation we remove the sentence " Figure 1a and 1b illustrate these results." and we refer to Figures 1a and 1b in the text.

Point 6: Why is the mean and its standard error represented instead of the standard deviation? The authors should justify this analysis.

Response: The standard deviation (SD) is a measure of dispersion of the raw data points. The use of the standard error (SE) instead of the standard deviation is very common in the scientific literature. In fact, both statistical parameters are related by the expression: SE=SD/

Point 7: To support and confirm the lipid peroxidation results, an assay for ROS determination in hepatocytes should be performed.

Response: Lipid peroxidation is a process under which oxidants such as free radicals attack lipids especially polyunsaturated fatty acids (PUFAs). This reaction produces a wide variety of oxidation products including many different aldehydes. Among them, malondialdehyde (MDA) have been extensively studied by Esterbauer and his colleagues in the 80s [https://doi.org/10.1016/0891-5849(91)90192-6]. For this reason, we have chosen malondialdehyde and 4-hydroxyalkenals for estimation of lipid peroxidation. The biochemical assays for these molecules are not particularly sensitive but they are widely used in the scientific literature.

Point 8 and 9: Immunohistochemical assays with specific markers against T-cells (CD3, CD4, CD8), macrophages (like IBA-1) and neutrophils should be performed. What about glycogen synthesis in this tissue? PAS staining should be included to analyze PAS-positive liver cells.

Response: Ischemia–reperfusion is a very complex phenomenon characterized by a severe metabolic imbalance between supply and demand which causes partial or complete loss of hepatic function. Therefore, a large number of measurements could be performed, including studies of tissue-infiltrating leukocyte markers and the effect on hepatic glycogen metabolism. Undoubtedly, these measurements may provide more mechanisms involved in ischemia reperfusion injury. Our study focuses on assessing the effects of hepatic ischemia reperfusion on biological membranes and whether the antioxidant melatonin can protect them. For this purpose, we studied in cellular and mitochondrial membranes, a functional parameter, such as their fluidity, and two biochemical indicators of oxidative stress of lipids and proteins, which are the main constituents of biological membranes. The study was completed with two morphological indicators of lipid and protein oxidative damage by immunohistochemistry. We are sorry, we couldn’t do further assays to detect other indicators of hepatic damage due to Ischemia reperfusion.

Point 10: Figure 4 is very poorly displayed; it should be included at higher magnification to observe the immunostaining because it is not very conclusive in this way. In addition, it also does not show details such as leukocytic infiltration and increased lipid peroxidation in the sinusoids.

Response: We have presented figure 4 at a higher magnification and to observe the immunostaining it is displayed vertically. In relation to the leukocytic infiltration, we have marked with red arrows in the figure (b and f) several lymphocyte aggregates. The increase in the labeling of anti-MDA (Figure 4c) is refered at the level of cytoplasm and membranes of the hepatocytes that are close to the centrolobular vein (lines 289-90)

Point 11: All events highlighted by the authors in section 4 should also be marked with arrows or other graphic elements.

Response: We have added several marks in the figure and in its legend. The main changes of ischemia (b and f) were hepatic vacuolization (red asterisks*), and leukocyte infiltration (red arrows), whereas the reperfusion indicated mainly hyperchromatic hepatocyte nuclei (red arrows) and oxidative damage (DAB+) (red arrowheads) specially to lipids (c) and to a lesser degree to proteins (g), in the proximity of the centrolobular vein (c and g), and in the hepatic cords (c).

Round 2

Reviewer 1 Report

I am satisfied with the responses provided by the authors addressing my comments and concerns. The manuscript has now been significantly improved.

Author Response

Thank you very much for your advice and guidance. Your review has greatly helped us improve our manuscript.